# Remote electrophysiological cardiotocography (eCTG), evaluation of feasibility in complicated pregnancies from 32 until 37 weeks gestational age in a home@hospital setting (HASTA): A prospective cohort study protocol

Sofie van Weelden[1,2,3]*, Loes Monen[1,2,3], M. Beatrijs van der Hout-van der Jagt[1,2,3,4], Judith O. E. H. van Laar[1,2,3]

1 Department of Gynecology and Obstetrics, Máxima MC, Veldhoven, The Netherlands, 2 Department of Electrical Engineering, Eindhoven University of Technology, Eindhoven, The Netherlands, 3 Eindhoven MedTech Innovation Centrer, Eindhoven, The Netherlands, 4 Department of Biomedical Engineering, Eindhoven University of Technology, Eindhoven, The Netherlands

* s.vanweelden@mmc.nl

## Abstract

### Introduction

High-risk pregnancies, including those complicated by pre-eclampsia, fetal growth restriction or preterm pre-labor rupture of membranes, often require hospitalization for fetal and maternal monitoring. Home monitoring may reduce the psychological and family burden, improve patient satisfaction, and lower health care costs. The primary diagnostic tool for fetal home monitoring is cardiotocography (CTG). However, conventional CTG can suffer from poor signal quality due to maternal obesity or movements. Since electrophysiological cardiotocography (eCTG) relies on electrical activity in cardiac and uterine muscles, signal quality is less impacted by obesity and motion artefacts than with conventional CTG, making eCTG also interesting for home monitoring. As remote eCTG is self-administered, it eliminates the need for professional support at home, potentially increasing autonomy and satisfaction. Nevertheless, signal quality of self-administered eCTG in the preterm period is unknown. This prospective cohort study will assess the feasibility of self-administered eCTG, using NemoRemote in a hospital setting between 32–37 weeks of gestation, simulating home use.

### Methods

In this single center prospective cohort study, 60 pregnant patients (≥18 years) with a singleton high-risk pregnancy from 32 until 37 weeks of gestational age will be recruited. Remote self-administered eCTG will be performed either daily for patients admitted to the hospital, or at least twice a week at the outpatient clinic. The primary

**Data availability statement:** The study is currently ongoing, and data collection and analysis have not yet been completed. Therefore, no (anonymized) dataset is available at this stage. Upon completion of the study and finalization of the dataset, the minimal anonymized dataset necessary to replicate the study findings will be uploaded to a stable, public repository in accordance with PLOS ONE's data availability policy. The relevant URLs, DOIs, or accession numbers will be provided at that time.

**Funding:** No peer-reviewed external funding was received for this study. This work was supported by University Fund Eindhoven, by Máxima Fund, and by the PPP Allowance TKI HTSM, made available by Top Sector Holland High Tech to the University Fund Eindhoven to stimulate public–private partnerships (grant PPS23-2-03529539). Manufacturer Nemo Healthcare B.V. loaned two NemoRemote devices for use in this study at no cost.

**Competing interests:** The authors have declared that no competing interests exist.

outcome is the proportion of successful eCTG recordings. Secondary outcomes include maternal and perinatal outcomes, patient and healthcare professional satisfaction, and a cost analysis.

## Conclusion

This study will provide valuable insights into the technical and logistical feasibility of self-administered eCTG monitoring in high-risk pregnancies, with potential implications for the implementation of remote fetal monitoring. A hospital-based feasibility study was chosen over an implementation study to safely assess signal quality of remote eCTG before broader application.

## Trial registration

This trial is registered on the Dutch trial register "Onderzoeksportaal" (NO. NL-OMON57084) https://onderzoekmetmensen.nl/en/trial/57084. Date registered: 05/11/2024 and the international trial register "ClinicalTrials.gov" (NO. NCT06859177); http://clinicaltrials.gov/study/NCT06859177. Date registered: 05/03/2025

## Introduction

### Background

Cardiotocography (CTG) is recommended in international guidelines for monitoring maternal and fetal well-being in complicated pregnancies [1–3]. However, its often leads to frequent outpatient visits or hospital admissions, even for otherwise healthy women [1–4]. Antenatal admissions can cause psychological stress due to separation from home, limited mobility, and uncertainty [5]. In general, remote care offers a promising alternative, potentially improving access to care, enhancing patient satisfaction while reducing health care costs due to a reduction in visits and admissions [6,7]. CTG telemonitoring is a relatively new approach in complicated pregnancies and is recognized as an alternative to hospital admission [6–9]. Several telemonitoring platforms for remote CTG have been assessed in prospective studies, demonstrating technical and logistical feasibility as well as acceptance by both patients and clinicians [5]. Nonetheless, conventional CTG equipment faces certain limitations when applied in telemonitoring contexts.

The conventional CTG signals that are generated by Doppler ultrasound and tocodynamometry probes are prone to artefacts and signal loss due to fetal and maternal movements, particularly in patients with a high body mass index [10,11]. With an estimated global prevalence of maternal obesity of 20.9% (95%CI; 18.6–23.1) in 2024, expected to increase to 23.3% (95%CI; 20.3–26.2) by 2030, this is a common issue [12]. In case of poor signal quality, the transducer often has to be repositioned repetitively [13], which complicates self-administered home monitoring. Moreover,

it might be difficult for women to locate the fetal heart rate (FHR) with the Doppler ultrasound probe, potentially causing maternal stress. Additionally, patients might experience the tightly fitted elastic bands as uncomfortable and it substantially limits their mobility [13].

Due to the limitations of conventional CTG monitoring, extensive research has been carried out to develop new, non-invasive, monitoring methods. Electrophysiological CTG (eCTG) monitoring derives the CTG signal from electrodes placed on the maternal abdomen. It monitors FHR by fetal electrocardiography, maternal heartrate by maternal electro-cardiography, and the electrical activity of the uterine muscle by electro hysterography [14–17]. The electrophysiological recordings are hardly affected by maternal or fetal movement or abdominal wall thickness, as they are transferred through conduction of electrical signals across the underlying tissues [11,18]. Furthermore, the wireless setup allows women to move freely [13]. However, signal quality of self-administered eCTG in the preterm period is unknown.To assess this, non-invasive, self-monitoring method for applicability in a home setting in late preterm complicated pregnancies, we will conduct a prospective cohort study as described in this protocol. Hospital setting was defined as fetal monitoring at the obstetric ward and/or at the outpatient clinic.

## Objectives

This study will evaluate the technical and logistical feasibility of remote eCTG, using NemoRemote (NRM) in a hospital setting between 32 and 37 weeks of gestational age (GA), simulating conditions as if the patient was at home. Secondary objectives of this study are to assess maternal and perinatal outcomes, patients and healthcare professionals' (HCPs) satisfaction, and costs.

## Methods and analysis

This HASTA study will be conducted as a single center prospective cohort study between 26 March 2025 and 1 April 2026, aiming to include 60 eligible patients between 32 and 37 weeks of GA. The study will take place at the obstetric department of the Máxima MC (MMC), which provides both secondary and tertiary maternity care. The aim of this study is to assess the feasibility of home-based monitoring in a hospital setting, where patients will self-apply the device as they would at home.

This study protocol (S1 Appendix) and execution of the study will be fully compliant to the most recently updated version of the Declaration of Helsinki (64th version, 2013). The researchers will protect confidential data in accordance to the rules and regulations stated in the Dutch General Data Protection Regulation. This study adheres with the Standard Protocol Items: Recommendations for Interventional Trials (SPIRIT) guidelines (S2 Appendix).

The project was awarded in 2023, and the protocol version number is 3.0, dated 4 November 2024. Research activities commenced on 26 March 2025, and are expected to continue through 1 April 2026. As of 15 September 2025, a total of 23 participants had been enrolled. Preliminary findings are anticipated to be available in early 2027. Study findings will be published in peer-reviewed journals and shared in lay language with participants who opt in for feedback. These results are expected to inform future implementation of remote maternal and fetal monitoring in high-risk pregnancies between 32 and 37 weeks of gestation. A detailed schedule (SPIRIT) of enrolment, interventions and assessments for the HASTA study is presented in Fig 1.

Fig 2 shows the NemoRemote Monitoring (NRM) system (Nemo Healthcare) used in this study, which is CE-certified for clinical use. The NRM Information for Users (IFU) is available on request at Nemo Healthcare B.V. (https://nemohealth-care.com/en/contact/). NRM is an electrophysiological recording device including a Nemo application, that non-invasively measures FHR, uterine activity and maternal heart rate from the fetal and maternal electrocardiography and electrohys-terography, which are acquired from abdominal surface electrodes. NRM consists of a link charger (base), a link, and an abdominal patch with six electrodes. Recorded data will be transferred between NRM link and the tablet via a wireless

| TIMEPOINT | Enrolment | Monitoring | | | | | | Close-out |
|---|---|---|---|---|---|---|---|---|
| | $-t_1$ 31+6 to 36+5 weeks of gestational age | $t_0$ 32 to 36+6 weeks of gestational age | $t_1$ Day after start eCTG | $t_2$ End of fetal monitoring, ≤37+0 weeks of gestational age | $t_3$ Child birth | $t_4$ 4 weeks after childbirth | $t_5$ 6 weeks after childbirth | $t_6$ End of study, 6 weeks after childbirth last participant |
| **ENROLMENT** | | | | | | | | |
| Eligibility screen | X | | | | | | | |
| Informed consent | X | | | | | | | |
| Allocation | | X | | | | | | |
| **INTERVENTIONS** | | | | | | | | |
| Self-administered remote eCTG | | X | X | | | | | |
| Self-measured vital functions and symptoms | | X | | | | | | |
| Questionnaire patients | | X | X | X | | X | | |
| Questionnaire healthcare professionals | | | | | | | | X |
| **ASSESSMENTS** | | | | | | | | |
| Baseline characteristics | X | | | | | | | |
| eCTG | | X | X | | | | | |
| Maternal and perinatal outcomes | | X | X | X | X | X | X | |
| Well-being patient (EQ-5D5L/VAS) | | | X | | | X | | |
| Experience patient eCTG (D-QUEST) | | | | X | | | | |
| Satisfaction patient received care (CSQ-8) | | | | | | X | | |
| Satisfaction healthcare professionals provided care (D-QUEST) | | | | | | | | X |
| Costs | | X | X | X | | | | |
| (Serious) Adverse Events | | | X | X | X | X | X | |

**Fig 1. The SPIRIT schedule of enrolment, interventions and assessments for the HASTA study.**

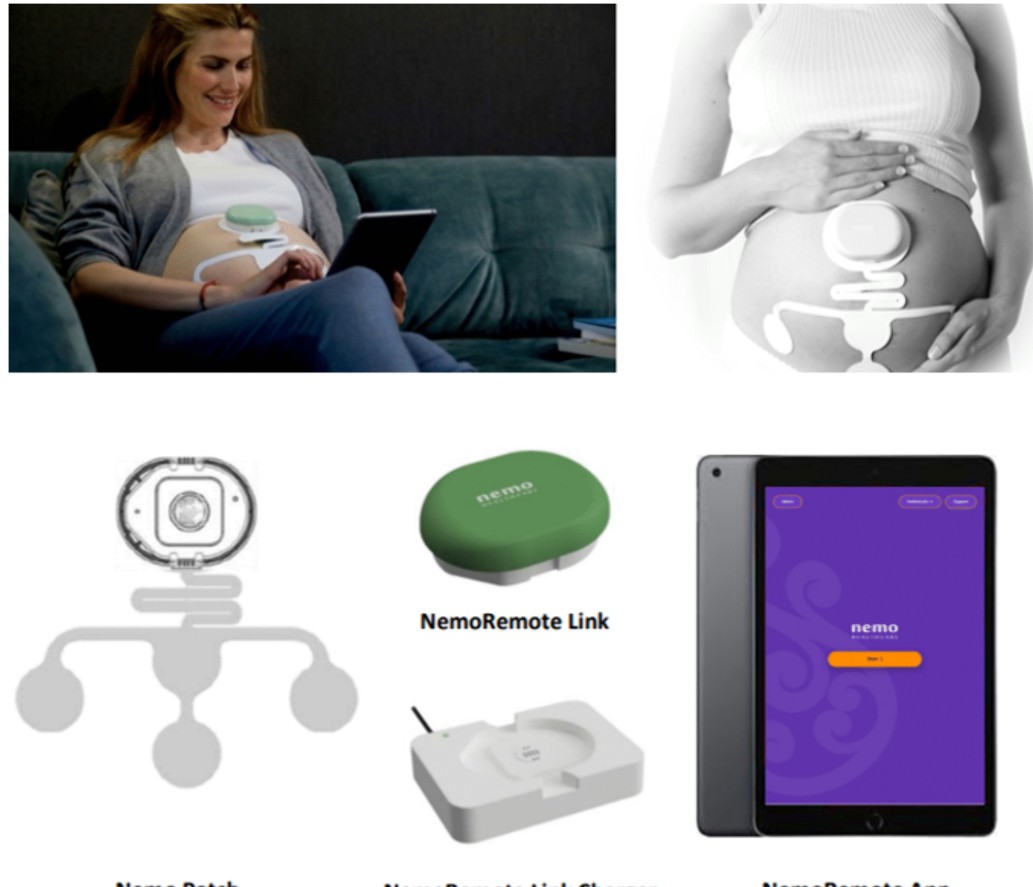

**Fig 2. The NRM used in the HASTA study consists of three components: charger (base), link and patch.** Nemo Healthcare B.V. (source: NemoRemote Monitor® Information for Users).

connection. Simultaneous measurements with conventional CTG will not be performed, and clinicians will not be blinded to the mode of acquisition. The purpose of this study is to evaluate workflow feasibility and signal quality of eCTG in routine care, rather than to validate eCTG against conventional CTG.

## Sample size

This study aims to include 60 patients, sample size calculation is based on expert's calculation using data from Quéméré et al [19], evaluating home monitoring using CTG. In that study, 4% of all recordings required repetition and 0.9% of the CTGs needed to be repeated in hospital. The initial success rate was 95.5% increasing to 99%, after one repeat recording. Similar to current clinical policy used for CTG, this study will allow a technically insufficient but clinical reassuring remote eCTG recording to be repeated once on the same day to still be considered successful. For all inclusions, a success rate of ≥90% within two attempts is regarded clinically acceptable based on prior studies [20–22] and expert opinion. To ensure a lower confidence limit of at least 82.5% (Wilson method, 95% CI), 60 participants are required to assess feasibility based on signal quality and success of remote eCTG recording the first day (S3 Appendix). Máxima MC conducts approximately 2,400 deliveries annually and operates as a regional center for high-risk pregnancies.

## Population

To be eligible in this study, patients must be 18 years or older, with a singleton pregnancy and with a GA from 32 + 0 up to 36 + 6 weeks. Additionally, eligible patients must have an indication for fetal monitoring at least twice per week (e.g., pre-eclampsia (PE), fetal growth restriction (FGR) and preterm pre-labor rupture of membranes (PPROM)), with absence of exclusion criteria at least 24 hours after admission and oral and written informed consent must be obtained.

Exclusion criteria for participation in this study are: an indication for intravenous medication; a blood pressure >160/110 mmHg; absent-/or reversed flow in the umbilical artery Doppler measurement; HELLP syndrome; obstetric intervention expected within 48 hours; patients admitted with a clinical diagnosis of sepsis with hypotension; insufficient knowledge of Dutch or English language; insufficient comprehension of instructions or patient information; fetal and/or maternal cardiac arrhythmias; contraindications to abdominal patch placement as described in the NRM user information; patients connected to an external or implanted electrical stimulator, such as Transcutaneous Electro Neuro Stimulation or pacemaker. Inclusion and exclusion criteria are described in Table 1. Pregnancy-related diagnoses, relevant for this study, are clarified in Table 2.

**Table 1. HASTA study inclusion and exclusion criteria.**

| Inclusion criteria | Exclusion criteria |
|---|---|
| Pregnant with a singleton pregnancy, ≥ 18 years | Maternal: indication for intravenous medication, blood pressure >160/110 mmHg, HELLP syndrome (Table 2), clinical diagnosis of sepsis with hypotension (i.e., septic shock), cardiac arrhythmias. |
| Gestational age from 32 + 0 up to 36 + 6 weeks | Fetal: absent-/or reversed flow umbilical artery Doppler, cardiac arrhythmias. |
| Any indication for fetal monitoring at least twice per week (e.g., pre-eclampsia, fetal growth restriction and preterm pre-labor rupture of membranes) | Obstetric intervention expected within 48 hours (such as non-reassuring cardiotocography (CTG) active vaginal blood loss, signs of abruption placentae, meconium stained amniotic fluid, signs of chorioamnionitis) |
| Absence of exclusion criteria > 24 hours after admission | Insufficient knowledge of Dutch or English language and/or Insufficient comprehension of instruction NemoRemote or patient information |
| Oral and written informed consent is obtained | Contraindications to abdominal patch placement (dermatologic diseases of the abdomen precluding preparation of the abdomen with abrasive paper) |
| Maternal age ≥ 18 years | Patients connected to an external or implanted electrical stimulator, such as Transcutaneous Electro Neuro Stimulation and pacemaker (because of disturbance of the electrophysiological signal) |

**Table 2. Definitions of HASTA study relevant pregnancy-related diagnosis.**

**Pre-eclampsia (PE)**

Pre-eclampsia is pregnancy induced hypertension accompanied by one or more of the following new-onset conditions at or after 20 weeks' gestation [23]:
1. Proteinuria
2. Other maternal organ dysfunction, including:
    Acute kidney injury (creatinine ≥90 µmol/L; 1 mg/dL); Liver involvement (elevated transaminases, e.g., alanine aminotransferase or aspartate aminotransferase > 40 IU/L) with or without right upper quadrant or epigastric abdominal pain); neurological complications (examples include eclampsia, altered mental status, blindness, stroke, clonus, severe headaches, persistent visual scotomata); hematological complications (thrombocytopenia – platelet count below 150.000/µL, disseminated intravascular coagulation, hemolysis)
3. Utero-placental dysfunction (such as fetal growth restriction, abnormal umbilical artery Doppler wave form analysis, or stillbirth).

**Fetal growth restriction (FGR)**

Estimated fetal weight or abdominal circumference below the 3rd percentile, or below the 10th percentile combined with Doppler abnormalities (e.g., Umbilical Artery PI > 95th percentile or Cerebroplacental Ratio <5th percentile) [1].

**Preterm pre-labor rupture of membranes (PPROM)**

Rupture of membranes before gestational age of 37 weeks without contractions [8].

**HELLP syndrome (H**aemolysis, **E**levated **L**iver enzymes, and **L**ow **P**latelets)

The combination of all or some of haemolysis elevated liver enzymes and thrombocytopenia [23].
Combination of symptoms that signifies a more serious manifestation of PE. Hemolysis (Lactate Dehydrogenase ≥600 U/L, haptoglobin < 0.2g/L) AND elevated liver enzymes (alanine aminotransferase or alanine aminotransferase > 70U/L) AND Low platelets (<100*109/L).

## Study procedures

Eligible patients will be identified at admission or at the outpatient clinic if monitoring is indicated at least twice a week. These patients will be consecutively informed about the study by their obstetric caregiver and will receive a Patient Information Form. With permission, they will be contacted by a trained obstetric healthcare professional (e.g., resident, physician-assistant, (research) midwife or nurse) for further information. Questions will be addressed, and counselling will be provided by a researcher or trained delegate. If still eligible on the second day of admission, informed consent will be obtained. Due to the high-risk nature of the study population, which is often threatened with imminent delivery, the standard 7 days reflection period is shortened. Patients may request additional time to consider their decision. Participants will receive routine care as advised in national or local guidelines. Based on their clinical pathway, as shown in Fig 3, they will receive self-administered remote eCTG monitoring by NRM: at least once a day in-hospital or at least twice a week during outpatient follow-up. Recordings will be instantly transferred digitally to the electronic medical record (HiX 6.3, HF109.5) for clinical assessment. If recordings are reassuring but of insufficient quality, the recording will be repeated the same day, similar as in clinical practice for conventional CTG. If the eCTG signal quality is insufficient for clinicians to assess fetal condition, the protocol specifies switching to conventional CTG and/or performing an ultrasound assessment. No additional feedback mechanism is required because, similar to conventional CTG, eCTG provides a live trace with alarms linked to standard CTG interpretation criteria. Clinicians will interpret eCTG traces according to standard care without applying additional criteria. Criteria for successful recording are defined in a standard operating procedure, accessible for all trained HCPs involved. During the study, participants will self-monitor and log maternal vital signs and symptoms using the Safe@Home app (Luscii Healthtech), shown in Fig 4, simulating home use. Vital signs measured are heartrate, blood pressure, temperature and fluid balance. Symptom tracking covers preeclampsia-related-complaints, abdominal pain, contractions, blood loss, and fetal movements. As long as remote eCTG registrations are successful, participants will continue the study interventions until discharge, delivery or when 37 weeks is reached. At any moment– when requested by the participants or by the HCP– a switch to conventional CTG can be made. Patients who decide not to participate will receive standard care, which consists of conventional CTG monitoring, where measurements will be done by the HCP. In that case, patients will no longer register their measurements and symptoms in the Luscii® app.

Participants will receive automated emails from the electronic Case Report Form program (Research Manager) at inclusion, end of monitoring and four weeks postpartum. A single reminder will be sent after one week if the questionnaire is not completed. From inclusion up to six weeks postpartum, relevant clinical data of mother and child will be collected, as defined

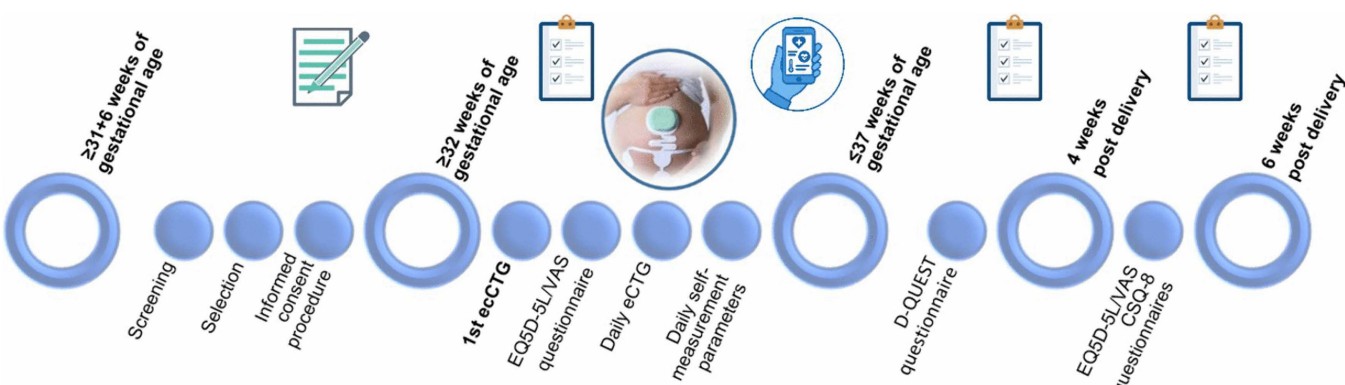

**Fig 3. Diagram – timeline of enrollment, interventions, assessment for participants.** CSQ-8, client satisfaction questionnaire – 8 items, D-QUEST: Dutch version of the Quebec User Evaluation of Satisfaction with assistive Technology, eCTG: electrophysiological cardiotocography, EQ5D-5L: EuroQol 5 dimensions and 5 levels, VAS: Visual Analogue Scale.

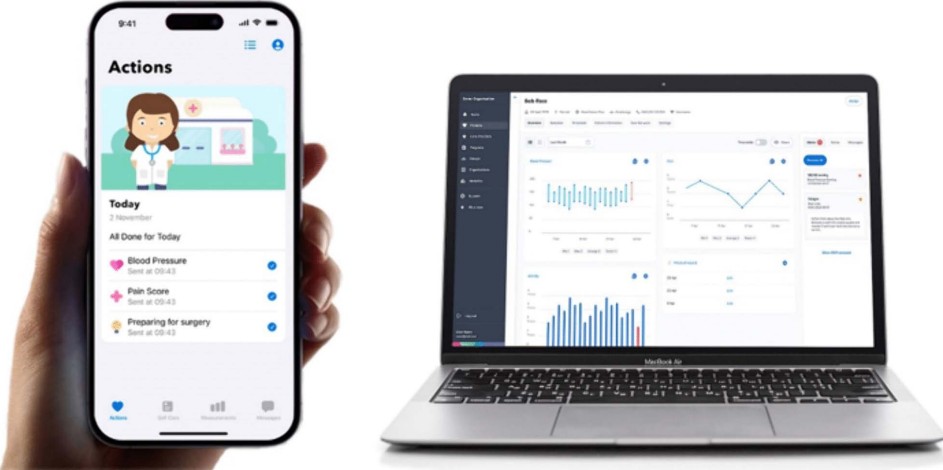

**Fig 4. The Safe@Home application used in the HASTA study (Luscii Healthtech B.V.).**

in outcome measures and in the study protocol (S1 Appendix). Additionally, this system is also used to send a questionnaire to all HCPs involved in this study once the last participant has either delivered or has reached a GA of 37 weeks.

## Outcome measures

**Primary outcome measures.** The primary outcome is the proportion of successful remote eCTG recordings, defined as at least 30 minutes of recordings with a continuous FHR signal for > 80% of the recording time [24], and judged by the HCP as sufficiently interpretable. If the initial trace is reassuring but not fully interpretable, due to signal loss, recording may be extended up to 90 minutes. To prevent early closure of the recordings, these criteria are outlined in a standard operating procedure accessible to all trained HCPs. After study enrollment, all recordings will be exported for objective assessment and analysis of the primary outcome. The success of eCTG will be evaluated retrospectively and objectively, based on exported eCTG data. This approach ensures that the study focuses on feasibility and usability rather than comparative diagnostic accuracy.

**Secondary outcome measures.** Secondary outcome measures include eCTG parameters, maternal, perinatal and neonatal parameters, patient well-being and satisfaction, HCP satisfaction and cost analysis.

**eCTG outcomes** include the number of days of remote monitoring and number of remote eCTG recordings per participant. Additional eCTG outcomes include amount of repeated remote recordings on the same day due to reassuring but insufficient traces, and switches to conventional CTG due to clinical indications or poor signal. The success rate of these conventional recordings will be assessed using the same criteria as in remote eCTG monitoring. The incidence of skin irritation or other (serious) adverse events will be recorded.

**Maternal outcomes** including maternal baseline characteristics (GA at inclusion, diagnosis and reason for monitoring, duration from inclusion to delivery), maternal morbidity (emergency/secondary Cesarean section, postpartum hemorrhage > 1000mL, abruption placentae, eclampsia, HELLP (Table 1), umbilical cord prolapse, pulmonary or deep venous thrombosis, and chorioamnionitis) and mortality will be assessed. Maternal mortality is defined as death from any cause related to or aggravated by pregnancy or its management (excluding accidental or incidental causes) occurring during pregnancy, childbirth, or within 42 days after termination of pregnancy [25,26].

**Perinatal and neonatal outcomes**, including baseline characteristics (GA at birth, birth weight), morbidity (medium care unit and/or neonatal intensive care unit admission, dysmaturity (<3rd or <10th percentile), congenital anomalies, signs

of asphyxia (Apgar score <7 at 5 minutes and/or umbilical cord pH<7.05)), and mortality will be assessed. Perinatal mortality is defined as the number of fetal deaths past 22 completed weeks of gestation (154 days) plus the number of deaths among live-born children up to seven completed days of life [25,26], while neonatal mortality refers to deaths occurring between the 7th day and the 28th day of life (day 7–27) [25,26].

**Well-being and satisfaction** of all participating patients will be assessed through the EQ-5D-5L, which includes several validated questionnaires like Visual Analogue Scale (VAS) [27], D-QUEST [28] and CSQ-8 [29]. The HCP satisfaction concerning the monitoring method, will be assessed using the D-QUEST questionnaire [28], adapted for professional use with an introductory explanation of modifications to clarify interpretation from a HCPs perspective.

**Costs** of home monitoring versus in-hospital monitoring will be calculated under the assumption that home monitoring using eCTG has already been implemented, compared to in- hospital care using conventional CTG. Clinical pathways follow standard clinical practice, independent of study participation.

## Data management and safety considerations

Study outcomes will be obtained from the electronic medical record and questionnaires. In case of missing data, additional information will be requested from the participant's primary care midwife, child health clinic, general practitioner, pediatrician, and/or gynecologist affiliated with the referral hospital. The data will be pseudonymized and the decryption key will only be available in Máxima MC for members of the research team. Research manager® (ISO 27001 certified) is used to build this database. The method for data storage and handling has been approved by the data security officer of Máxima MC. The Medical Ethical Committee Máxima MC, Veldhoven, The Netherlands, extensively reviewed the study protocol (version 3) and concluded that the study is in line with the Medical Research Involving Human Subjects Act (S1 Appendix). Therefore, ethical approval was granted (Trial reference numbers: W24.066, NL87858.015.24, NL-OMON57084). Máxima MC Board of Management also approved the conduct of this study at their hospital (2024.0137). Important modifications of the study protocol will be communicated with relevant parties (e.g., Medical Ethical Committee, trial register, and researchers). Written informed consent will be obtained from all subjects involved in this study (S4 Appendix).

Monitoring will be performed in compliance with Good Clinical Practice to achieve high-quality research and secure patient safety. The Clinical Trial Center Maastricht will independently monitor the conduct of this study. Visits will be scheduled prior to study initiation, after the inclusion of the first inclusions and upon study closure. In accordance to section 10, subsection 4, of the WMO, the sponsor will suspend the study if there is sufficient ground that continuation of the study will jeopardise subject health or safety. The sponsor will notify the accredited METC without undue delay of a temporary halt including the reason for such an action. The study will be suspended pending a further positive decision by the accredited METC. The investigator will take care that all subjects are kept informed.

## Statistical analysis

**Primary data** will be analyzed using SPSS version 29 (IBM Corp., Armonk, NY, USA) and SAS software version 9.4 (SAS Institute Inc., Cary, NC, USA). Signal quality will be defined as the percentage of recorded time where there is no FHR signal loss. Signal quality will be determined for each patient individually. Mean, SD, and 95% confidence interval (CI) of the mean success rate will be calculated. Missing data will be evaluated and reported as counts and percentage (N, %). A limited amount of missing data is expected due to the data collection approach.

**Secondary data** will be analyzed using SPSS (version 29). Baseline demographics, medical characteristics, and secondary outcomes will be summarized. Normally distributed data will be reported as mean±standard deviation; non-normal distribution as median with interquartile range. Sensitivity analyses subgroup analyses (e.g., PE, FGR, PPROM) are planned. The EQ-5D-5L (including EQ VAS) [27], D-QUEST [28] and CSQ-8 [29] exploratory questionnaires will be analyzed and summarized as described below. Results will be reported as mean (SD), minimum, median, and maximum.

- **EQ-5D-5L:** This assessment tool assigns a numerical value to each response level (i.e., 1 for "no problems", 5 for "extreme problems"/"unable to") and the summing of these values across the 5 items, results in a score. The score is then placed on a scale, which is numbered from 0 to 100, respectively ranging from the worst to best health one can imagine. Results at inclusion and at 4 weeks postpartum will be separately reported.

- **EQ VAS:** The Visual Analogue Scale (0–100) will be summarized, with results at inclusion and 4 weeks postpartum separately reported. This assessment tool assigns a score between 0 and 100 – indicating their overall health will be summarized. A higher score meaning a better health.

- **D-QUEST:** This questionnaire comprises 12 items that can influence the users' satisfaction of a device and the provision process with a 5-point rating system and 2 composite items with a 5-point rating system. Total score ranges from 14 to 70, with the higher number indicating greater satisfaction. 5-point rating: not satisfied at all (1), displeased (2), more or less satisfied (3), satisfied (4) or very satisfied (5). Personal additional information can be given in a free description space. Patients are asked to underline the 3 most important aspects of the questionnaire. Free description will be imported into Atlas.ti for qualitative analysis and coded by a single researcher (SW) into four predefined categories: advantages and disadvantages of the NRM. Illustrative quotes will be translated into English and presented in the results.

- **CSQ-8:** The Client Satisfaction Questionnaire consists of 8 items scored from 1 to 4, with items 2, 4, 5, and 8 reverse scored. Total scores range from 8 to 32, with higher scores indicating greater satisfaction.

  **Cost analysis** will consist of an estimate of the value for money afforded once fetal home monitoring, using eCTG, is implemented. This hypothetical home monitoring clinical pathway (as if the women were at home), is compared to the actual conventional in-hospital monitoring clinical pathway (ward/outpatient clinic). Provided healthcare will be converted into a cost estimation by multiplying the number of healthcare units used (e.g., in-hospital/home admission day, (e)CTG measurement, outpatient clinic visits, transport by ambulance), by standard unit prices. All results will be reported as mean (SD;95%CI), median (IQR), number and percentage (N, %). A sub-analysis will compare clinical pathways between women requiring a hospital admission and those indicated of at the outpatient monitoring at the time of inclusion.

## Ethics and dissemination

Participation in this study is expected not to cause any risk for the patient or fetus. In case the interpretability of remote eCTG registration is deemed insufficient, a switch to the conventional CTG will be made. Remote eCTG monitoring with NRM patch offers mobility due to its wireless design, and self-application enhances women's autonomy. Skin irritation may occur with the use of the NRM patch, especially in combination with skin preparation (IFU). In such cases, reducing patch use and adjusting skin preparation can support recovery, as outlined in the NRM user information (IFU). If skin irritation occurs within 12 hours following patch removal, the investigator (SW) will record this adverse event in the patient's medical record and the electronic case report form. The event will also be reported in the study results. Additionally, a follow-up will be conducted to assess the persistence of these or any other adverse events. Serious adverse events, including maternal, fetal, and neonatal mortality, will be reported to the ethics committee and registered in the national trial register (CCMO).Liability insurance policy has been obtained by MMC (policy number 626.107.173, Centramed). Insurance coverage for subjects participating in this research is provided in accordance with the legal requirements of article 7 of the WMO (policy number 624.100.045, Centramed).

## Discussion

This prospective cohort study will provide important data on the technical and logistical feasibility of self-monitored electro-physiological CTG monitoring in high-risk pregnancies, between 32 and 37 weeks of gestation, with potential implications for remote eCTG implementation. As eCTG is primarily used in-hospital, its suitability of remote application within the

preterm period remains unexplored. A hospital-based feasibility study was selected to safely and efficiently assess signal quality and data transfer of remote eCTG in a controlled clinical setting. Additionally, various components of remote eCTG monitoring, such as both patient's and HCP's experience, maternal and perinatal outcomes, and costs will be explored, allowing potential challenges to surface before broader implementation.

Several limitations must be acknowledged. Monitoring in a clinical setting may influence patient-reported outcomes, such as perceived well-being and satisfaction. The EQ-5D-5L questionnaire (including VAS) [27], will be administered at inclusion and at four weeks postpartum to assess well-being at two distinct time points. The questionnaires are included to gain a broader understanding of the general well-being of the study population, rather than to measure any expected effect of the intervention, as no measurable effect is anticipated. Self-monitoring in a hospital may deter participation due to added responsibilities and lack of home comfort, though it may also enhance a sense of autonomy among participants.

Since the primary objective of this study is to assess technical reliability prior to safe implementation, sample size calculation is based on the success rate of eCTG recordings. This study is not designed to assess the safety of home monitoring. As the study focus is on technical feasibility, no comparison group is included, and therefore a formal cost-effectiveness analysis is not feasible. Instead, we will descriptively compare current clinical data to the anticipated outcomes in future home-based care. Any operational challenges that may arise at the start of the study, such as integration with the electronic health record system, will be systematically recorded and evaluated as part of the feasibility assessment.

No substantial changes over time are anticipated, as the expected inclusion period for this study is relatively short (within one year). If any substantial changes in clinical practice occur, a note to file will be added to the Investigator Site File and descripted in the discussion section of the final study report.

## Supporting information

**S1 Appendix. Study protocol.**
(PDF)

**S2 Appendix. SPIRIT checklist.**
(PDF)

**S3 Appendix. Sample size calculation.**
(TIF)

**S4 Appendix. Informed consent form.**
(PDF)

## Acknowledgments

Prof. E.R. van den Heuvel, statistician Eindhoven MedTech Innovation Center Mathematics and Computer Science (Statistics) provided expert support in calculating the required sample size for this study. Nemo Healthcare has no role in the study design, data collection, analysis, decision to publish, or preparation of the manuscript.

## Author contributions

**Conceptualization:** Sofie van Weelden, Loes Monen, M. Beatrijs van der Hout-van der Jagt, Judith O. E. H. van Laar.

**Project administration:** Sofie van Weelden.

**Supervision:** Loes Monen, M. Beatrijs van der Hout-van der Jagt, Judith O. E. H. van Laar.

**Visualization:** Sofie van Weelden.

**Writing – original draft:** Sofie van Weelden, Loes Monen, M. Beatrijs van der Hout-van der Jagt, Judith O. E. H. van Laar.

**Writing – review & editing:** Sofie van Weelden, Loes Monen, M. Beatrijs van der Hout-van der Jagt, Judith O. E. H. van Laar.

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
