## [Decision Letter · Decision Letter 0]

3 Nov 2025

Dear Dr. van Weelden,

Thank you for submitting your manuscript to PLOS ONE. After careful consideration, we feel that it has merit but does not fully meet PLOS ONE’s publication criteria as it currently stands. Therefore, we invite you to submit a revised version of the manuscript that addresses the points raised during the review process.

**ACADEMIC EDITOR: Please respond to all reviewers comments**

We look forward to receiving your revised manuscript.

Kind regards,

Ahmed Mohamed Maged, MD

Academic Editor

PLOS ONE

Journal Requirements:

“No peer reviewed  external funding was received for this study. The authors disclosed receipt of the following financial support for the research, authorship, and/or publication of this article. This work was supported by University Fund Eindhoven, by Máxima Fund and by the PPP Allowance TKI HTSM, made available by Top Sector Holland High Tech to the University Fund Eindhoven to stimulate public–private partnerships, grant number PPS23-2-03529539. Manufacturer Nemo Healthcare B.V. has loaned two NemoRemote devices for use in this study at no costs.”

4. We note that the original protocol that you have uploaded as a Supporting Information file contains an institutional logo. As this logo is likely copyrighted, we ask that you please remove it from this file and upload an updated version upon resubmission.

Reviewers' comments:

Reviewer's Responses to Questions

**Comments to the Author**

1. Does the manuscript provide a valid rationale for the proposed study, with clearly identified and justified research questions?

Reviewer #1: Yes

Reviewer #2: Yes

Reviewer #3: Yes

Reviewer #4: Yes

Reviewer #5: Yes

2. Is the protocol technically sound and planned in a manner that will lead to a meaningful outcome and allow testing the stated hypotheses?

Reviewer #1: Yes

Reviewer #2: Yes

Reviewer #3: Yes

Reviewer #4: Yes

Reviewer #5: Yes

3. Is the methodology feasible and described in sufficient detail to allow the work to be replicable?

Reviewer #1: Yes

Reviewer #2: Yes

Reviewer #3: Yes

Reviewer #4: Yes

Reviewer #5: Yes

4. Have the authors described where all data underlying the findings will be made available when the study is complete?

Reviewer #1: Yes

Reviewer #2: Yes

Reviewer #3: Yes

Reviewer #4: Yes

Reviewer #5: Yes

5. Is the manuscript presented in an intelligible fashion and written in standard English?

Reviewer #1: Yes

Reviewer #2: Yes

Reviewer #3: Yes

Reviewer #4: Yes

Reviewer #5: Yes

You may also provide optional suggestions and comments to authors that they might find helpful in planning their study.

Reviewer #1: As the statistical reviewer I will focus on methods and reporting. the power calculations are appropriate.

Major

1) For the primary outcome an one sample t-test is proposed. against what hypothesised mean? I would advise against heavily relying on p-value based tests and statistical significance considering the sample.

2) how will missing data be handled, or none are expected?

3) how will the EQ-5D and other questionnaires be analysed, as the authors say?

Minor

1) Will changes over time be assessed and how?

Reviewer #2: This study protocol describes a prospective, single-center feasibility study evaluating self-administered remote electrophysiological cardiotocography (eCTG) in high-risk pregnancies between 32 and 37 weeks’ gestation. The concept is highly topical and clinically relevant, given the growing interest in telemonitoring and patient-autonomous obstetric care.

1. The operationalization of “interpretability” is subjective. To increase reproducibility, specify objective quantitative thresholds (e.g., continuous FHR signal >80 % of recording time; uterine activity identifiable in >90 % epochs) and give detail training/standardization of assessors.

2. The calculation (n = 60) is based on feasibility proportions from a 1990s home-CTG study, not on modern eCTG signal behavior. The authors should provide power considerations for the primary outcome (≥90 % successful recordings, 95 % CI ≥ 82.5 %), with an explicit formula or simulation demonstrating how 60 subjects suffice to assess signal reliability.

3. Describe how eCTG results will be validated against conventional CTG during monitoring. Are clinicians blinded to the mode of acquisition when interpreting the trace? Outline a feedback mechanism if discrepancies arise (e.g., when eCTG signal is reassuring but Doppler CTG shows decelerations).

4. The manuscript lists comprehensive inclusion/exclusion criteria but should elaborate on criteria for early termination in case of persistent signal failure or maternal stress.

5. Consider adding an adverse-event monitoring flow diagram showing reporting hierarchy (participant → site investigator → METC).

Reviewer #3: Dear Authors - It was a privilege to review your manuscript – “Remote electrophysiological cardiotocography (eCTG), evaluation of feasibility in complicated pregnancies from 32 until 37 weeks gestational age in a home@hospital setting (HASTA): A prospective cohort study protocol.”

This is a well-conceived, clinically relevant feasibility protocol addressing a pressing implementation gap: whether self-administered electrophysiological CTG (eCTG) can deliver interpretable fetal monitoring in late preterm, high-risk pregnancies. The rationale is strong (motion/BMI robustness; autonomy; potential cost relief), the target population is appropriate (32–37 weeks; pre-eclampsia/FGR/PPROM), and the design prudently simulates home use within a monitored hospital environment. With some clarifications, especially around the definition of outcomes, this well-thought-out protocol will provide decision-ready evidence to inform remote fetal monitoring pathways. Integrating the study with an electronic health record system for extraction of maternal health data and transmission of results for reading and interpretation will underpin case identification, monitoring workflows, and outcomes ascertainment in the study.

General comment: HASTA - Thank you for providing the clarification on the acronym HASTA. For a global audience, it will be helpful to provide clarification of what it stands for in this context of this study.

Short Title: Feasibility remote eCTG monitoing home@hospital (Feasibility [of] remote… [missing word])

(.. eCTG monitoing [correct the typo – “monitoring”])

Abstract: Home monitoring may reduce the psychological and family burden, improve patient satisfaction and lower health care costs. (Revision – add a comma “,” between separated items)

Abstract: signal quality is hardly affected by obesity and motion than conventional CTG, (This sentence is unclear due to its grammatical structure. The use of “hardly” and “than” in a comparative expression is generally incorrect. Consider replacing “hardly” with “less” or other appropriate phrasing for a global audience.)

Abstract - Methods: “Remote self-administered eCTG will be performed daily during hospital admission, or at least twice a week at the outpatient clinic.” (The description of the study design is not clear – the cohort design is explained to be a hospital-based study, while in the methods, there is a mention of outpatient clinics [which is not generally considered a hospital setting except for hospital-based outpatient clinics]. Clearly define what will be considered a “hospital setting” in the context of this study. For ease of referencing, you should define, for the study, what is considered a hospital setting.)

Abstract/Methods: Definition of the primary outcomes (Consider reviewing and clarifying the parameters of the primary outcome. The sentence in this section seems contradictory – “successful eCTG recordings, defined as … due to signal loss”.

Furthermore, in Line 191, you define a different set of parameters for the primary outcomes. )

Keywords: Second Trimester (This should be corrected to “third” trimester – the cohort group is GA 32 – 37 weeks)

Line 170 – (Correct “in-“ to the complete word “inclusion” for standardization for Table 2 header)

Table 2 (In your description of HELLP Syndrome key punctuations, commas (“,”), to distinguish each component of the HELLP syndrome as separate items.)

General review comments - (The overall use of future tenses in some places and past or present tenses was noticed in some places. There are confusing “verb tense” reconciliations. Consider making the scenario and objectives clear in the abstract and introduction and maintain the consistency of the verb-tense agreements throughout the manuscript. Careful attention must be paid to appropriately reflecting ongoing activities versus proposed activities. For example, Line 122-123: “The study will be conducted…” vs Line 123-124 – “The study is conducted…”"

Line 191-194 (Include the SOP as an appendix and insert the appropriate reference to indicate it)

Line 277 (The initial introduction of an acronym must be completely outlined before independently using the acronym subsequently - consider lettering out “Physical Examination” upon this initial introduction.)

Line 336-338 (Correct the sentence for clarity. “…will [be] systematically recorded and evaluated as part of the feasibility assessment.)

Line 268 (Consider replacing “electronic patient file” with a more standardized nomenclature – “electronic medical record”.)

Reviewer #4: Congratulations on your scientific work. This is an engaging and relevant topic that will significantly contribute to enhancing the evaluation and follow-up of pregnant women requiring fetal well-being monitoring. We look forward to the forthcoming results, which will undoubtedly have a meaningful impact on the treatment costs of these patients within the healthcare system.

Reviewer #5: Overall Summary

This manuscript presents a very interesting study with a neat design for a prospective cohort. The construction and conceptual framework are robust, and the study's aim has the potential to lead to a significant breakthrough in fetal wellbeing monitoring. While the foundation is strong, several minor issues require addressing before publication can be recommended.

Major Comments

Sample Size Justification: The proposed sample size of 60 subjects is ambitious given the epidemiology of the conditions under study. The prevalence of severe preeclampsia is approximately 2%, and fetal growth restriction is only slightly higher. To recruit a sufficient number of cases, the study likely requires a facility with an annual delivery rate of approximately 3,600. The authors should explicitly state this as a key parameter in their sample size calculation to clarify the feasibility and generalizability of their recruitment strategy.

Definition of Fetal Growth Restriction (FGR): The diagnostic criteria for FGR provided in Table 2 are vague and do not reflect current clinical standards. The definition used aligns more closely with Small for Gestational Age (SGA), which is a distinct entity with different management and prognostic implications (Chen & Li, 2023, Ultrasound in Obstetrics & Gynecology). Contemporary guidelines, such as those reviewed by Giouleka et al. (2023, Obstetrical & Gynecological Survey), define FGR as an estimated fetal weight below the 3rd percentile, or below the 10th percentile combined with Doppler abnormalities (e.g., Umbilical Artery PI >95th percentile or Cerebroplacental Ratio <5th percentile). Adopting this updated, precise definition is critical for accurate patient stratification and interpretation of results.

Minor Comments

Methodological Transparency: The timeline for the study and the process for sequential inclusion, exclusion, and dropout are clearly described. The proposed statistical analysis plan is sound and appropriate for the study design.

Declaration of Commercial Instrument: The study utilizes a specific, proprietary device (the NEMO healthcare system). As this may introduce a systematic instrumental bias and only one brand currently offers this technology, this relationship should be explicitly disclosed in a conflict of interest statement to ensure full transparency and address any potential ethical considerations.

Recommendation

Minor Revisions.

This is a promising paper with a strong foundational design. The concerns regarding sample size justification and the definition of FGR are critical to address, but with these revisions, the manuscript will be significantly strengthened and suitable for publication.

**Do you want your identity to be public for this peer review?** For information about this choice, including consent withdrawal, please see our Privacy Policy

Reviewer #1: No

Reviewer #2: **Yes:** Burak Bayraktar

Reviewer #3: **Yes:** Babajide Adewumi

Reviewer #4: **Yes:** JOAO FELIX DIAS

Reviewer #5: **Yes:** Juan Carlos Bello-Muñoz

---

## [Author Response · Author response to Decision Letter 1]

30 Dec 2025

EDITOR, POINT 1

COMMENT: Please state what role the funders took in the study. If the funders had no role, please state: "The funders had no role in study design, data collection and analysis, decision to publish, or preparation of the manuscript.”

RESPONSE: We thank the editor for pointing out the need to clarify the role of the funders in the study. In response, we have added a statement in the Funding section specifying that the funders had no role in study design, data collection and analysis, decision to publish, or preparation of the manuscript.

Funding: “Manufacturer Nemo Healthcare B.V. has loaned two NemoRemote devices for use in this study at no costs.” Has now been replaced by: “For the duration of the study, NemoRemote devices have been provided free of charge by Nemo Healthcare B.V. The funders and Nemo healthcare had no role in study design, data collection and analysis, decision to publish, or preparation of the manuscript.” This is also filled out at the PLOS ONE authors submission website – Funding section.

EDITOR, POINT 2

COMMENT: We note that you have indicated that there are restrictions to data sharing for this study. For studies involving human research participant data or other sensitive data, we encourage authors to share de-identified or anonymized data. However, when data cannot be publicly shared for ethical reasons, we allow authors to make their data sets available upon request. Before we proceed with your manuscript, please address the following prompts:

If there are ethical or legal restrictions on sharing a de-identified data set, please explain them in detail (e.g., data contain potentially identifying or sensitive patient information, data are owned by a third-party organization, etc.) and who has imposed them (e.g., a Research Ethics Committee or Institutional Review Board, etc.).

If there are no restrictions, please upload the minimal anonymized data set necessary to replicate your study findings to a stable, public repository and provide us with the relevant URLs, DOIs, or accession numbers.

Please update your Data Availability statement in the submission form accordingly.

RESPONSE: Thank you for your comment. Our study is still ongoing, and therefore no anonymized dataset is currently available for upload. As the data collection and analysis have not yet been completed, it is not possible to provide the minimal dataset required to replicate the study findings at this stage.

We confirm that once the study is completed and the dataset is finalized, we will upload the minimal anonymized dataset to a stable, public repository and provide the relevant URLs, DOIs, or accession numbers in accordance with PLOS ONE’s data availability policy.

We have added the following section to the Data Availability Statement: “The study is currently ongoing, and data collection and analysis have not yet been completed. Therefore, no (anonymized) dataset is available at this stage. Upon completion of the study and finalization of the dataset, the minimal anonymized dataset necessary to replicate the study findings will be uploaded to a stable, public repository in accordance with PLOS ONE’s data availability policy. The relevant URLs, DOIs, or accession numbers will be provided at that time.”

EDITOR, POINT 3

COMMENT: Please also provide contact information for a data access committee, ethics committee, or other institutional body to which data requests may be sent.

RESPONSE: We thank the editor for highlighting the need to provide contact information for data access. In response, we have added the details of the appropriate institutional body to which data requests can be directed in the Data Availability section of the manuscript.

Corresponding author: Weelden, Sofie van S.vanWeelden@mmc.nl

Ethics committee: METC METC@mmc.nl

Principal Investigator: Laar, Judith van judith.van.laar@mmc.nl

Data requests: Sofie van Weelden

EDITOR, POINT 4

COMMENT: We note that the original protocol that you have uploaded as a Supporting Information file contains an institutional logo. As this logo is likely copyrighted, we ask that you please remove it from this file and upload an updated version upon resubmission.

RESPONSE: We thank the reviewer for noting that our protocol contains an institutional logo. We have made the following changes:

D-QUEST logo has been removed from the questionnaire in the appendix of the protocol.

• Study Protocol: page 50.

REVIEWER 1, POINT 1

COMMENT: For the primary outcome an one sample t-test is proposed. against what hypothesized mean? I would advise against heavily relying on p-value based tests and statistical significance considering the sample.

RESPONSE: We thank the reviewer for noting that our protocol proposed a one-sample t-test for the primary outcome. We agree that this test is not suitable, as no comparison between two groups will be performed. We have carefully reconsidered this point and provided a detailed description in the study protocol to ensure reproducibility. The phrases “using a one-sample t-test”, and “In case of non-normally distributed data, the Kruskal-Wallis test will be used as one-way ANOVA analog”, has been removed from the protocol paper, as rightly suggested by the reviewer.

Accordingly, the reference to a t-test has been removed from the study protocol. A note to file has been added to the Investigator Site File, and this adjustment will be included in a subsequent amendment. We have removed the phrase “using a one-sample t-test” and “In case of non-normally distributed data, the Kruskal-Wallis test will be used as one-way ANOVA analog”, from the study protocol and protocol article.

• Method - Statistical analysis: page 10, line 320.

• Study protocol: page 39, section 9.1 Primary study parameters.

REVIEWER 1, POINT 2

COMMENT: How will missing data be handled, or none are expected?

RESPONSE: We thank the reviewer for raising this point regarding missing data. We would like to clarify that, due to the data collection approach, a limited amount of missing data is expected.

We kindly refer you to: “In case of missing data, additional information will be requested from the participant’s primary care midwife, child health clinic, general practitioner, pediatrician, and/or gynecologist affiliated with the referral hospital.”

• Method - Data management: page 8, lines 241-243.

We have added the following sentence to the manuscript: “Missing data will be evaluated and reported as counts and percentage (N, %). A limited amount of missing data is expected due to the data collection approach.”

• Method – Missing data: page 11, lines 320-321.

REVIEWER 1, POINT 3

COMMENT: Questionnaires: how will the EQ-5D and other questionnaires be analyzed, as the authors say?

RESPONSE: We have now provided a detailed description of the analytical approach used for questionnaire data.

We have clarified the analysis of the questionnaires in the following section:

• Method - Statistical analysis – Secondary data: page 11, lines 329-349.

REVIEWER 1, POINT 4

COMMENT: Will changes over time be assessed and how?

RESPONSE: No substantial changes over time are anticipated, as the expected inclusion period for this study is relatively short (within one year). If any substantial changes in clinical practice occur, a note to file will be added to the Investigator Site File and descripted in the discussion section of the study results article. Important modifications of the study protocol will be communicated with relevant parties (e.g., Medical Ethical Committee, trial register, and researchers) (data management: page 9, lines 254-256). We have added the following lines to the manuscript:

“No substantial changes over time are anticipated, as the expected inclusion period for this study is relatively short (within one year). If any substantial changes in clinical practice occur, a note to file will be added to the Investigator Site File and descripted in the discussion section of the final study report.”

• Discussion: page 12, line 388-390.

REVIEWER 2, POINT 1

COMMENT: The operationalization of “interpretability” is subjective. To increase reproducibility, specify objective quantitative thresholds (e.g., continuous FHR signal >80 % of recording time; uterine activity identifiable in >90 % epochs) and give detail training/standardization of assessors.

RESPONSE: We agree with the reviewer that interpretability, as judged by healthcare professionals, is inherently subjective and may complicate reproducibility of the analysis. Therefore a standard operating procedure (SOP) was written and healthcare professionals will be trained, primarily to prevent early closure (<30 minutes) of the eCTG recordings. Considering that all recordings will be exported post-enrollment for objective evaluation and analysis of the primary outcome, we concluded that inclusion of the SOP would not yield additional benefit. The standard text used in the medical record, as outlined in the SOP, is presented below.

We have clarified the procedure in the following lines: “After study enrollment, all recordings will be exported for objective assessment and analysis of the primary outcome.”

• Outcomes Measures – Primary outcomes measures: page 8, lines 202-203.

We have added: “The success of eCTG will be evaluated retrospectively and objectively based on exported eCTG data. This approach ensures that the study focuses on feasibility and usability rather than comparative diagnostic accuracy.”

• Outcome Measures – Primary outcome measures: page 8, lines 203-205.

The standard text in the medical record, as outlined in the SOP, includes the following:

• eCTG: (xxx) bpm, variability (present/not present/reduced/acceptable), accelerations (present/absent), decelerations (present/absent). Additional comment: (No suspicion of fetal hypoxia / consistent with gestational age / differential diagnosis: umbilical cord compression).

• Recording: minimum (yes/no) 30 min, (more/less) than 20% signal loss.

• Assessment: (normal/suboptimal/abnormal).

• eCTG repeated: (No/Yes, due to reassuring but insufficient recording / due to other reason:).

• Switch: (no/yes,) (temporary/permanent) (if yes, reason for switch:/) (signal quality/other reason:/).(*Switch due to signal quality, possible cause:/ Switch due to other reason:/)

REVIEWER 2, POINT 2

COMMENT: The calculation (n = 60) is based on feasibility proportions from a 1990s home-CTG study, not on modern eCTG signal behavior. The authors should provide power considerations for the primary outcome (≥90 % successful recordings, 95 % CI ≥ 82.5 %), with an explicit formula or simulation demonstrating how 60 subjects suffice to assess signal reliability.

RESPONSE: In response to the reviewer’s concern regarding power considerations, we would like to clarify that the sample size calculation was based on a desired success probability of 90%, informed by local healthcare professionals’ opinions on clinical acceptability and supported by previous literature. To ensure a reliable estimate of the success rate, the calculation was performed by Prof. E. van den Heuvel, a statistical expert at Eindhoven University of Technology.

We aimed for the lower bound of the one-sided 95% confidence interval to be at least 82.5%. Using Wilson’s method for proportions, the required sample size was determined by:

Where P=0.90, Z=1.645 (one-sided 95% CI), and Δ is the maximum allowable distance between the estimate and the lower confidence limit. For a sample of 39 participants and a success probability of 90%, the 95% lower limit is approximately 80%. To achieve a lower limit of at least 82.5%, 60 participants are required (Wilson method, 95% CI). Therefore, we chose a sample size of 60 to provide additional data for secondary outcomes while ensuring the primary outcome meets the precision requirement.

To clarify the sample size calculation we have added the following:“….sample size calculation is based on expert’s calculation using data from….”

• Method - Sample Size: page 5, lines 141-142.

We have added the sample size figure in the appendix of the manuscript as Supporting Information.

• Method - Sample Size: (S2 Appendix), page 6, line 150.

• Supporting information: S2 Appendix. Figure Sample Size calculation, page 12, line 398.

REVIEWER 2, POINT 3

COMMENT: Describe how eCTG results will be validated against conventional CTG during monitoring. Are clinicians blinded to the mode of acquisition when interpreting the trace? Outline a feedback mechanism if discrepancies arise (e.g., when eCTG signal is reassuring but Doppler CTG shows decelerations).

RESPONSE: We thank the reviewer for raising this point. We would like to clarify that simultaneous measurements are not performed, and clinicians are not blinded to the mode of acquisition. The NEMO Remote Monitoring (NRM) system is already CE-certified; therefore, this study does not aim to compare eCTG with conventional CTG. If the eCTG signal quality is insufficient for clinicians to assess fetal condition, we consider this a result from self-applied monitoring, rather than from eCTG quality itself. Yet, to mitigate situations where CTG is not available due to problems with self-monitoring, the protocol specifies switching to conventional CTG and/or performing an ultrasound assessment. A separate feedback mechanism is not required because, similar to conventional CTG, eCTG provides a live trace with alarms linked to standard CTG criteria. Clinicians interpret the traces according to standard care without applying additional criteria. The success of the eCTG recordings will be evaluated retrospectively and objectively based on exported eCTG data. We have added to following sections to the manuscript:

“The study will utilize the NEMO Remote Monitoring (NRM) system, which is CE-certified for clinical use.”

• Method - Design and Setting: page 4, lines 124-127.

“Simultaneous measurements with conventional CTG will not be performed, and clinicians will not be blinded to the mode of acquisition. The purpose of this study is to evaluate workflow feasibility and signal quality of eCTG in routine care, rather than to validate eCTG against conventional CTG.”

• Method - Design and Setting: page 5, lines 131-134.

“The success of eCTG will be evaluated retrospectively and objectively, based on exported eCTG data. This approach ensures that the study focuses on feasibility and usability rather than comparative diagnostic accuracy.”

• Method – Outcome Measures – Primary outcome measures: page 8, lines 203-205.

“If the eCTG signal quality is insufficient for clinicians to assess fetal condition, the protocol specifies switching to conventional CTG and/or performing an ultrasound assessment according to clinical decision. No additional feedback mechanism is required because, similar to conventional CTG, eCTG provides a live trace with alarms linked to standard CTG interpretation criteria. Clinicians will interpret eCTG traces according to standard care without applying additional criteria.”

• Method – Procedure: page 9, lines 279-285.

REVIEWER 2, POINT 4

COMMENT: The manuscript lists comprehensive inclusion/exclusion criteria but should elaborate on criteria for early termination in case of persistent signal failure or maternal stress.

RESPONSE: As specified in the original study protocol (Procedures section), we have addressed this point by adding the following clarification to the manuscript:

“As long as remote eCTG registrations are successful, participants will continue the study interventions until discharge, delivery or when 37 weeks is reached. At any moment –– when requested by the participants or by the HCP – a switch to conventional CTG can be made. Patients who decide not to participate will receive standard care which consists of conventional CTG monitoring, where measurements will be done by the HCP. In that case, patients will no longer register their measurements and symptoms in the Luscii® app.”

• Method - Procedures: page 9, line 290-295.

REVIEWER 2, POINT 5

COMMENT: Consider adding an adverse-event monitoring flow diagram showing reporting hierarchy (participant → site investig

---

## [Decision Letter · Decision Letter 1]

9 Jan 2026

Remote electrophysiological cardiotocography (eCTG), evaluation of feasibility in complicated pregnancies from 32 until 37 weeks gestational age in a home@hospital setting (HASTA): A prospective cohort study protocol.

PONE-D-25-50846R1

Dear Dr. van Weelden,

We’re pleased to inform you that your manuscript has been judged scientifically suitable for publication and will be formally accepted for publication once it meets all outstanding technical requirements.

Kind regards,

Ahmed Mohamed Maged, MD

Academic Editor

PLOS One

Additional Editor Comments (optional):

Reviewers' comments:

Reviewer's Responses to Questions

**Comments to the Author**

1. Does the manuscript provide a valid rationale for the proposed study, with clearly identified and justified research questions?

Reviewer #1: Yes

2. Is the protocol technically sound and planned in a manner that will lead to a meaningful outcome and allow testing the stated hypotheses?

Reviewer #1: Yes

3. Is the methodology feasible and described in sufficient detail to allow the work to be replicable?

Reviewer #1: Yes

4. Have the authors described where all data underlying the findings will be made available when the study is complete?

Reviewer #1: Yes

5. Is the manuscript presented in an intelligible fashion and written in standard English?

Reviewer #1: Yes

You may also provide optional suggestions and comments to authors that they might find helpful in planning their study.

Reviewer #1: I am satisfied with the authors' responses and the resulting changes to the paper. I have no further comments to make.

**Do you want your identity to be public for this peer review?** For information about this choice, including consent withdrawal, please see our Privacy Policy

Reviewer #1: No

---

## [Editor Report · Acceptance letter]

PONE-D-25-50846R1

PLOS One

Dear Dr. van Weelden,

I'm pleased to inform you that your manuscript has been deemed suitable for publication in PLOS One. Congratulations! Your manuscript is now being handed over to our production team.

Kind regards,

on behalf of

Professor Ahmed Mohamed Maged

Academic Editor

PLOS One